# Method Validation and Characterization of the Associated Uncertainty for Malondialdehyde Quantification in Exhaled Breath Condensate

**DOI:** 10.3390/antiox10111661

**Published:** 2021-10-22

**Authors:** Maud Hemmendinger, Jean-Jacques Sauvain, Nancy B. Hopf, Pascal Wild, Guillaume Suárez, Irina Guseva Canu

**Affiliations:** 1Center for Primary Care and Public Health (Unisanté), Department of Occupational and Environmental Health, University of Lausanne, 1066 Lausanne, Switzerland; jean-jacques.sauvain@unisante.ch (J.-J.S.); nancy.hopf@unisante.ch (N.B.H.); guillaume.suarez@unisante.ch (G.S.); irina.guseva-canu@unisante.ch (I.G.C.); 2National Research and Safety Institute (INRS), 54500 Vandœuvre-lès-Nancy, France; Pascal.Wild@inrs.fr

**Keywords:** malondialdehyde, oxidative stress biomarkers, uncertainty, exhaled breath condensate

## Abstract

There are several methods for quantifying malondialdehyde (MDA), an oxidative stress biomarker, in exhaled breath condensate (EBC). However, due to the very diluted nature of this biological matrix, a high variability is observed at low concentrations. We aimed to optimize a 2,4-dinitrophenylhydrazine-based method using liquid chromatography coupled to tandem mass spectrometry and characterize the uncertainty associated with this method. We investigated the following parameters for the method validation: calibration linearity, limit of detection (LOD), precision, recovery, and matrix effect. The results were used to identify the main sources of uncertainty and calculating the combined uncertainty. The applicability of this method was evaluated in an ongoing epidemiological study by analyzing 164 EBC samples collected from different professional groups in subway environments. The optimized method was sensitive (LOD: 70 pg/mL), precise (inter-day variation < 19%) and accurate (recovery range: 92–106.5%). The calculated analytical uncertainty was the highest at the LOQ level and reached 23%. Although the analytical uncertainty was high at low MDA concentrations, it was significantly lower than that the observed inter-individual variability. Hence, this method performs sufficiently well and can be recommended for future use in epidemiological researches relying on between-subject differences.

## 1. Introduction

The burden of non-communicable chronic disease is increasing in the most developed, but is also increasing in developing countries [1]. Most of these diseases, including cardiovascular diseases, respiratory diseases, and cancer include oxidative stress and inflammation mechanism in the disease pathway. Oxidative stress corresponds to an imbalance between the presence of noxious oxidants like reactive oxygen species (ROS) and the endogenous defense mechanisms, in favor of the former [2]. ROS are quite reactive, persist only for a very short time in vivo and do not accumulate to levels high enough to be directly measured. ROS can oxidize proteins, lipids, and nucleic acids, causing structural and functional cellular changes. Therefore, the presence of ROS is quantified indirectly by measuring oxidized products, also known as oxidative stress biomarkers. Oxidized lipids and their metabolites have been proposed as biomarkers of oxidative stress [3], as lipids are concentrated in biological membranes and are vulnerable to ROS [4,5]. A great variety of lipid hydroperoxides is formed depending on the degree and mechanism of oxidation [6,7]. Malondialdehyde (MDA) is generated after lipid oxidation as a result of lipid hydroperoxides breakdown [8]. MDA is of particular interest due to its relative stability [9,10] and can be considered a potential candidate as an oxidative stress biomarker to evaluate lipid oxidation [11]. MDA is considered as the most commonly studied oxidative biomarker [8]. MDA levels have been found to be significantly increased in several respiratory diseases such as lung cancer [12], asthma [13], chronic obstructive pulmonary diseases [14,15], or occupational lung diseases (asbestosis, pleural hyalinosis, or silicosis) [16].

Measurements of biomarkers related to such lung diseases have involved invasive procedures including bronchial biopsy, induced sputum, and bronchoalveolar lavage [17]. Conversely, the collection of exhaled breath has recently emerged as an appealing non-invasive procedure to characterize the state of the lower respiratory airways [18], including by analyzing different markers in the liquid phase obtained by condensing exhaled breath (exhaled breath condensate, EBC) [19]. The main advantages are that EBC is a relatively simple matrix, is easy to collect, and gives direct access to the lung as a target organ [20]. However, there are some major analytical challenges hampering the use of EBC routinely. EBC mainly consists of condensed water (>99%) and biomarkers are highly diluted, resulting in typical concentrations at the pg/mL level. A limited amount of EBC sample volume (about 3 mL EBC collected in 20 min), and lack of standardization for the collection and concentration expression contribute to the disadvantages of this matrix [20].

Most of the methods used for MDA analysis in EBC include a derivatization step to increase MDA stability [21], resulting in improved chromatographic separation, mass spectrometer (MS) ionization and MS/MS fragmentation detectability [22,23]. Larstad and colleagues [24] were the first to describe the use of thiobarbituric acid (TBA) for MDA analysis in EBC by using liquid chromatography-fluorescence detection. Whereas harsh derivatization conditions were employed (>90 °C), they didn’t find any artifactual production of MDA from the auto-oxidation of lipids or from non-lipid-related materials, as often reported for other complex matrices [25]. Nevertheless, in order to be selective to MDA, the TBA reagent is not recommended [26] and separation combined with mass detection should be used instead. This latter detection has the advantage of defining specifically the mass of the desired compound rather than simply detecting color change. Other derivatizing reagents such as 2,4-dinitrophenylhydrazine (DNPH) are more specific for carbonyl compounds [27]. The chemical reaction using this derivatization agent is shown in Figure 1.

In addition, the MDA-DNPH complex produces distinct, easily detected peaks in chromatograms using HPLC with MS detection [28]. Although this derivatizing agent is an improvement, interferences are reported from atmospheric aldehydes entering during the processing or reagent impurities, leading to significantly measurable MDA in the method blanks [8,27,29]. This can lead to lack of accuracy and precision in the analytical methods. An estimation of the quality and the confidence of the results are crucial for establishing valid diagnostic tests before their clinical application. Nevertheless, the accuracy and precision of the analytical method could be assessed through measurement of uncertainty, corresponding to the statistical dispersion of the obtained MDA concentration values in EBC attributed to a measured quantity [30].

Our objectives were to (1) optimize the existing method of MDA analysis in EBC based on DNPH derivatization followed by high-pressure liquid chromatography with tandem mass spectrometry (HPLC-MS/MS); (2) validate this method and characterize its uncertainty, and (3) define the method’s applicability in epidemiological studies focusing on particulate exposure by analyzing EBC samples obtained from subway workers in a pilot field study.

## 2. Materials and Methods

### 2.1. Reagents-Chemicals

MDA-salt (MDA tetrabutyl ammonium salt) (96%, neat) was obtained from Sigma-Aldrich (St. Louis, MO, USA). MDA-d2 (1,1,3,3-tetraethoxypropane-d2, stock solution: 98%) was obtained from Cambridge Isotope Laboratories (Tewksbury, MA, USA). DNPH reagent (2,4-Dinitrophenylhydrazine) in a solid form containing 25% water, was obtained from Carlo Erba Reagents (Chaussée du Vexin, Val de Reuil, France). HPLC grade methanol (≥99.9%) was obtained from Merck (Buchs, Switzerland). LC-MS grade solvents, methanol (≥99.95%) and acetonitrile (≥99.9%) were obtained from Carlo Erba Reagents (Chaussée du Vexin, Val de Reuil, France). LC-MS grade acetic acid was obtained from Honeywell (Seelze, Germany). High purity water was produced in our laboratory with a MilliQ Advantage water purification system (18.2 MΩ·cm at 25 °C, <3 ppb total organic carbon; Merck, Schaffhouse, Switzerland).

### 2.2. Preparation of Standards and Procedural Blanks

A stock solution of MDA at 5.4 µg/mL was prepared by diluting a weighted mass of MDA tetrabutyl ammonium salt (7 mg) in 3 mL of MeOH and further diluting it by a factor of 100 with milliQ water. The MDA stock solution was stored at −80 °C for 8 months. To overcome unwanted variations during derivatization and analysis, MDA-d2 was used as an internal standard (IS). The MDA-d2 stock solution was prepared via acidic hydrolysis of the 1,1,3,3-tetraethoxypropane-d2 standard according to a previously published protocol [31]. Briefly, 1,1,3,3-tetraethoxypropane-d2 (50 mg) were poured into 28 mL of 0.02 M HCl and left for 2 h at room temperature. The resulting MDA-d2 stock solution at 7.9 mM in 0.02 M HCl was then stored at 4 °C. These different stock solutions were diluted in milliQ water to prepare daily working solutions of MDA (1 and 20 ng/mL) and IS (283.5 ng/mL). Calibration standards were obtained by diluting the working solutions to get final MDA concentrations of 74, 148, 370, 740, 1110, 1480, and 2220 pg/mL, with a constant IS concentration of 15 ng/mL. The concentration of the internal standard was selected to be of comparable signal intensity observed for the MDA calibration standards. The criteria for linearity was assessed by means of the coefficient of determination (R2), fixed at R2 > 0.99. Procedural blanks correspond to milliQ water, with IS at a concentration of 15 ng/mL.

### 2.3. EBC Samples and Quality Control (QC)

EBC samples used for validation and quality control (QC) were collected from 13 healthy non-smoking voluntary adult participants. The sample included nine women and four men. For EBC collection, we used the commercially available breath condenser (Turbo Deccs, Medivac, Parma, Italy). All collected samples were pooled, aliquoted in plastic tubes (1 mL) and stored at −80 °C until analysis. These samples were used during the method development and validation, particularly for investigating the matrix effect and the limit of detection (LOD) for MDA, as well as QC during sample analysis. An average concentration of these pooled samples was calculated over twelve independent measurements to determine a baseline concentration. Then, they were spiked with known MDA concentrations and used for QC. QC were prepared at final concentrations of 211, 349, 642, 1180, and 2258 pg/mL with IS at 15 ng/mL. Each analysis sequence for validation included a seven-point calibration curve in duplicate, five EBC QC samples in quintuplicate, two non-spiked EBC, and six procedural blanks (Appendix A).

In order to assess the suitability of the validated method, we used 164 EBC samples collected in an ongoing occupational field study [32]. This study included nine non-smoking healthy adults of both sexes working in the underground subway in Paris. Three workers from three occupational groups were included. EBCs were collected twice daily (before and after the working shift) over 10 days following the latest recommendations of the American Thoracic Society and the European Respiratory Society Task Force [33]. Food and drinks consumed within 3 h before EBC collection were recorded in a standardized form. None of the participants declared drinking coffee within the hour before EBC collection. The exhaled air was condensed at −10 °C during calm oral respiratory ventilation for 2 × 10 min, using the Turbo Deccs. EBC (2–3 mL) was collected and aliquoted immediately away from the sampling area on a clean table. Collected EBC aliquots were frozen at −20 °C, transported, and stored at −80 °C until analysis. Nine analytical sequences (one for each volunteer) including one calibration curve (seven levels), the EBC samples, six procedural blanks and two QC controls (low: 211 pg/mL and high: 2258 pg/mL) were analyzed.

### 2.4. MDA Derivatization with DNPH

The DNPH derivatizing solution was prepared at a concentration of 396 µg/mL in a H2O:ACN:acetic acid mixture (6:3.8:0.2 *v*/*v*, pH ≈ 3.2) and stored in the dark at room temperature. Water was added gradually at the last stage to avoid precipitation of DNPH due to its low solubility in aqueous solution. The MDA derivative was prepared by incubating 135 µL of the sample (including 10 µL of IS) with 50 µL DNPH 396 µg/mL for 2 h at 50 °C. We found that this condition was sufficient for a complete reaction between the MDA and DNPH present in excess (Appendix A). The resulting mixture was immediately analyzed via HPLC-MS/MS after cooling to reduce possible interferences originating from additional sample treatment.

### 2.5. HPLC–MS/MS Analyses

The target analytes were analyzed with an ultra-high pressure liquid chromatography (LC) system (Dionex Ultimate 3000) coupled with a Triple-Stage Quadrupole MS (TSQ Quantiva Thermo Scientific—Reinach, Switzerland). A C18 column (Zorbax Eclipse Plus 2.1 × 100 mm, 1.8 µm, Agilent, Morges, Switzerland) at 30 °C was used for separation. The injection volume was 20 µL and the solvent gradient (flow rate of 0.25 mL/min) combined eluent A (H2O with 0.1% acetic acid) and eluent B (MeOH/ACN 7:3 with 0.1% acetic acid). The following program was used: 100% A at 0 min, decreasing to 45% at 1.1 min, then to 35% A at 5 min, then to 10% A at 5.5 min until 7.5 min and increasing to 100% A at 8 min until 14 min.

The detection of MDA-DNPH was performed through a heated electrospray ionization (ESI) source operated in positive ion mode with the following parameters: spray voltage, 3700 V; ion transfer tube temperature, 390 °C; and vaporizer temperature, 350 °C. For MDA-DNPH, the transition *m*/*z* 235→159 was used for quantification, whereas the two other transitions *m*/*z* 235→143 and *m*/*z* 235→189 were used for confirmation; for MDA-d2-DNPH, the quantification transition was *m*/*z* 237→161. Chromatography Data System software (version 7.2.10, Thermo Scientific Dionex Chromeleon 7) was used for data acquisition and processing.

### 2.6. Method Validation and Estimation of Its Expanded Uncertainty

The validation of the optimized method was carried out on three different days by considering linearity, limit of detection (LOD) and quantification (LOQ), intra-day and inter-day precisions, recovery as a measure of accuracy, storage stability, and matrix effects as described in FDA/ICH guidelines [34]. Additional information on the calculation of these parameters is given in the Appendix A.

Some of these validation parameters were used to estimate the expanded uncertainty of this analytical method [30]. We adopted a pragmatic approach to identify the main elements of uncertainty, using the overall method performance. We considered three parameters as main contributors to uncertainty: the precision, recovery, and purity of the MDA-salt. Precision represents all the effects covered by the intermediate precision study. It takes into account the daily variability of the calibration, including the different volumetric measuring devices (flasks and pipettes) used during the investigation. The recovery provides an indication of the accuracy of the concentration effectively found and consequently is subject to a degree of uncertainty. Finally, the MDA tetrabutyl ammonium salt used in this study is not 100% pure because it contains inorganic salts.

The general relationship between the combined standard uncertainty (u_c_) of a given MDA concentration in EBC and the uncertainty of the independent parameters is defined by the following Equation (1):(1)$$u_{c}\;(y(x1,\;x2,\;x3))=\sqrt{\sum_{i=1,n}Ci^{2}u{\left(xi\right)}^{2}}$$
where y(x1, x2, x3) is the function of the three considered parameters: precision, recovery, and purity. *Ci* is a sensitivity coefficient evaluated as *Ci* = ∂y/∂*xi* the partial differential of y with respect to each parameter. *Ci* describes how the value of y varies with changes in the parameters. *u*(*xi*) corresponding to the uncertainty related to each parameter is expressed as a standard deviation assuming Gaussian distribution of the parameters. The final expanded uncertainty is expressed by multiplying u_c_ with a coverage factor of 2 (to have a β-expectation tolerance interval at 95%). An example of calculating the uncertainty at the LOQ can be found in the Appendix A.

### 2.7. Statistical Analysis

In order to define the best calibration curve, we used the least square linear regression with an application of 1/x^2^ weighting as recommended by Gu et al. [35]. Coefficients of variation (CV) were calculated from the standard deviation and mean values. Basic calculations were performed with the built-in statistical functions in Microsoft Excel version 2016, whereas ANOVA, *t*-test, and Tukey’s test calculations were performed with the R program (R version 4.0.2, 22 June 2020—“Taking off again”). For all the EBC samples with concentrations lower than the LOD, a value of LOD/2 was attributed for statistical analysis [36,37].

## 3. Results

### 3.1. LC-MS/MS Analysis

LC separation and MS detection of MDA-DNPH in EBC was optimized to meet the highest sensitivity and repeatability. Due to the high concentration of DNPH reactive in the sample, the LC mobile phase gradient was optimized to guarantee a sufficient separation between the DNPH reactive and the MDA-DNPH analyte and avoid signal suppression. The 6 min washing period with eluent A at the end of the gradient program avoided a carry-over between different samples and protecting the analytical LC-MS system from solid deposition [38]. After derivatizing with DNPH, the MDA-d2-DNPH retention time was identical to the MDA-DNPH and only the *m*/*z* +2 mass of 237 was detected. The suitability of using MDA-d2 as IS for MDA quantification was thus confirmed.

### 3.2. Optimization of the DNPH Derivatization

Optimization phases focused on the most complete formation of the desired hydrazine derivative. As protons play a significant role in the derivatization process (Figure 1), different acids (perchloric acid, formic acid or acetic acid) were tested on standard solutions as well as on EBC samples. Acetic acid had a significantly lower variation in blank values response compared with the other tested acids (one-way ANOVA followed by Tukey’s test, *p* < 0.05; Appendix A). These variations were probably due to unwanted secondary reactions occurring at low pH and under oxidizing conditions, such as with perchloric acid. The final concentration of acetic acid was adjusted at 2%, as we observed a lower procedural blank signal when increasing the concentration of water.

To establish the optimal ratio of DNPH required to get an efficient derivatization of MDA in the sample, increasing DNPH concentrations in H2O:ACN:acetic acid mixture (6:3.8:0.2 *v*/*v*) from 10 to 1980 µg/mL was applied to MDA standard solutions at 35 ng/mL. This latter concentration corresponds to the highest MDA concentration in EBC reported in previous studies [3,39,40]. An increase of the MDA-DNPH peak area was observed with increasing molar ratios. The largest peak area of the MDA-DNPH derivative was obtained at a molar ratio of DNPH:MDA of 1600, corresponding to a final DNPH concentration of 55 µg/mL (Appendix A). This ratio is high due to the kinetics of the reactions between DNPH and MDA with a first reaction instantaneously involving DNPH and one of an aldehyde function of MDA and a second slower reaction leading to the closing of the pyrazole cycle by H_2_O elimination. In addition, DNPH could react not only with MDA but also with other aldehydes and ketones present in EBC samples. Thus, a large excess of DNPH was necessary to drive the reaction towards the MDA-derivative complex. For greater DNPH concentrations, the signal slowly decreased, suggestive of a signal suppression. In a parallel test with MDA-d2, a similar curve was observed as shown in Appendix A, demonstrating a proper choice of ratio for DNPH.

The effect of temperature on the derivatization reaction was examined in the range of 20–50 °C. We originally selected the incubation temperature of 37 °C, as it corresponds to the physiological temperature and exhibited the highest HPLC-MSMS signals (Appendix A). However, we observed that in these conditions, the derivatization rate was difficult to keep constant, and thus selected the incubation temperature at 50 °C for 2 h.

Figure 1 shows a typical liquid chromatogram for a procedural blank as well as the lowest standard level and an EBC sample using the optimized conditions. For the procedural blank, the signal at 7.35 min was identified as MDA-DNPH, as both the quantification ion (*m*/*z* 159) and confirmation ions (*m*/*z* 143 and *m*/*z* 187) presented the same retention time as the standard MDA. This indicated that an MDA source was present in the reactives used for this analysis. The contribution of this procedural blank to the signal of the lowest MDA concentration (74 pg/mL) was relatively large and variable, ranging from 45% to a maximum of 90%, depending on the day (SD: 15%, *n* = 12). To decrease such interferences, we tried to purify the DNPH derivatization solution by using a liquid-liquid extraction following Mendoca et al. [41]. However, this approach was unsuccessful, attributed to the small polarity difference between DNPH and MDA-DNPH. Changing the type of glassware, tips and other crimp caps did not reduce this contamination. We also examined the use of butylhydroxytoluene ((BHT), 10 μL of 2% BHT solution in the samples), considered by many authors as essential to prevent oxidation reactions leading to artifactual production of MDA [16]. We didn’t observe any improvement with BHT. Lastly, we examined if there could be a crosstalk between the MDA-d2 and the standard in a situation where protons might exchange with both deuterium in MDA-d2 and induce an internal contamination. We did not observe any modification in procedural blank when 5 times more internal standard was injected into the solution (Appendix A). Nevertheless, we observed that a reduction of the amount of organic solvent in favor of water in the DNPH derivatization solution decreased significantly the signal in the procedural blank (one-way ANOVA *p* = 0.038, data not shown). We also observed that this blank signal increased by 162% by re-using a DNPH solution stored at 4 °C for 1 day (*n* = 8). These results highlight the need to introduce procedural blanks in the analytical method to control the variation of contamination between batches and to use freshly prepared DNPH solutions. As we could not eliminate this signal, the EBC results had to be corrected with this procedural blank.

### 3.3. Method Validation and Estimation of Uncertainty

Table 1 summarizes the performances of the validated method following the FDA/ICH guidelines. The calibration curve, corresponding to the plot of the ratio signal MDA-DNPH/MDA-d2-DNPH as a function of the concentration of the added MDA was linear in the defined concentration range (74–2220 pg/mL) with linear regression coefficients R2 > 0.995 for all series. The slope variability (*n* = 12) was 10.8%, whereas a higher variability of 66.1% was determined for the intercept, indicative of a potential strong effect of the blank on the calibration curve. Over the entire standard concentration range, the observed percentage bias of back-calculated MDA concentrations was between 11.1 ± 7.4% (for the lowest concentration) and 3.0 ± 4.5% (for the greatest concentration). The LOD estimated from the error on the intercept was 70 ± 36.5 pg/mL, corresponding to a LOQ of 211 pg/mL. The maximum acceptable deviation observed at this concentration was smaller than 20% of the LOQ, as proposed by the FDA/ICH guidelines (Figure 2). Recovery rates for QC ranged from 92.4 ± 13.0% to 93.5 ± 7.3% for low (211 pg/mL) and high (2258 pg/mL) MDA concentration in EBC, respectively. The corresponding repeatability was smaller than 20% for low concentration (211 pg/mL) and 15% for the greatest concentration (2258 pg/mL).

The relative error of back-calculated EBC concentrations, related to their targeted concentrations (accuracy) is shown in the form of a box-and-whisker plot in Figure 2. The error for each concentration was comprised between the acceptance limits (fixed at 20%), demonstrating the validity of the method for the considered concentration range. The acceptance limits of 20% was exceeded for only two results that can be considered as potential outliers at the concentration of 211.2 pg/mL and the concentration 349.3 pg/mL, respectively, confirming the LOQ of 211 pg/mL.

Matrix effect was examined by comparing the slope of the calibration curves in water and in spiked EBC, using an unpaired *t*-test. No statistically significant difference was observed between the slopes from calibration standard (2.4 × 10^−4^ ± 0.31 × 10^−4^) and the EBC samples (2.3 × 10^−4^ ± 0.31 × 10^−4^) (unpaired *t*-test *p* > 0.05, *n* = 12).

Standard solutions of MDA at 5.4 µg/mL and IS at 567 µg/mL as well as QC EBC samples were observed to be stable for at least 8 months at −80 °C, with an observed decreased concentration <4% after that storage duration. Once derivatized, the MDA-DNPH compound was stable for at least 48 h when stored in an auto-sampler at room temperature (23 °C).

### 3.4. Uncertainty

The contribution of the three identified components to the uncertainty was determined using the data generated during the method validation process for EBC QC samples (Figure 3). The precision (repeatability) was the largest contributor to the uncertainty, reaching an average value of about 5.2%.

Figure 4 presents the calculated expanded uncertainty for the different MDA concentration in EBC, using a coverage factor of 2. As expected, the lowest uncertainty (≤10%) was observed for MDA concentrations above 650 pg/mL, whereas it increased to 23% for MDA concentrations near the LOQ.

### 3.5. Levels of MDA in EBC of Healthy Adult Workers

Table 2 presents the concentrations of MDA in EBC of nine workers, split in three different occupations. For all the samples, 19% of MDA concentrations were below the LOD (70 pg/mL), 63% were included between the LOD and LOQ (211 pg/mL), and 18% were above the LOQ (Appendix A). Concentrations of MDA were relatively low, with a highest concentration measured at 886 pg/mL.

The intra-individual MDA variability was between 33–85%, whereas the inter-individual variability was slightly smaller (41–57%). These variabilities must be compared to the estimated analytical uncertainty of about 23% for low (≈200 pg/mL) MDA levels in EBC (Figure 4).

To assess the advantages of our improved analytical method for epidemiological studies, we assumed: (1) a study design where each subject is sampled once; (2) log-normal distributions for MDA concentrations in EBC (as observed in our dataset); (3) a CV of 30% (worst case scenario) corresponding to an intra-individual variance on the log scale of 38% (intra-subject variability of 50%), and a total variance of 19% (inter-subject variability of 50%). We found the analytical variability to be far lower than the inter-subject variability, and conclude that the proposed analytical method is able to detect relevant inter-subject differences in epidemiological studies.

## 4. Discussion

We estimated the uncertainty of an optimized DNPH-based method for trace analysis of MDA in EBC. Our most significant finding is that procedural blank subtraction contributes substantially to the determined analytical uncertainty. 

For epidemiological purpose, we optimized and validated a sensitive and accurate DNPH-based method to quantify MDA concentrations in EBC samples from subway workers. Data obtained during the optimization revealed the systematic presence of a signal corresponding to MDA-DNPH in the procedural blank. Many publications report the use of DNPH to derivatize MDA in EBC [13,14,28,29], but only Kartavenka and colleagues [29] mention the presence of MDA-DNPH interfering peaks in the procedural blank samples. Our procedural blanks were quite variable between the different experiments even though we worked under strict conditions. We identified the DNPH reactive solution at the origin of this blank signal. Our attempts to reduce the contamination of this solution by liquid–liquid extraction [27,41] did not reduce the blank signal. We observed that using as little organic solvent as possible and using acetic acid instead of strong acids to prepare the DNPH reactive improved the analysis. Finally, solutions of DNPH should be freshly prepared for every analysis, even if some authors mention a good stability of the DNPH over 1 week in the refrigerator [28]. We concluded that good optimization of the sample pre-analytic preparation is essential to reduce the contribution of the procedural blank to the MDA signal of the EBC sample.

The presence of an MDA-DNPH signal in the procedural blank deserves more considerations regarding the determination of a LOD. There are several possible conceptual methods to calculate a LOD. Depending on the definition chosen, the values can vary greatly, which makes it difficult to compare the results between studies [43]. Considering the current lack of consensus in the field, it is important to have a complete reporting of detection limits and contamination levels, especially where a significant blank signal is found. In the present study, the LOD was calculated based on three times the error at the origin divided by the slope of the calibration curve. This method can be considered as conservative but our LOD appears to be quite comparable with two other similar studies reported in the literature (Table 3).

The method accuracies and precision demonstrated satisfactory performances following FDA/ICH guidelines (Table 1) and were also quite comparable to other similar methods (Table 3). We used the MDA-d2 as IS in order to correct for potential ion suppression or contamination.

Some authors recommend methyl-MDA as IS due to its stability [44] and simplicity of synthesis compared to MDA-d2 [45]. However, the derivatization yield with DNPH for methyl-MDA is different in the same conditions [27]. It generates two chromatographic peaks whereas only a single peak is observed with the MDA-d2. This phenomenon may be due to the formation of geometric isomers during the reaction between the methyl group with 2,4-DNPH [46]. Therefore, MDA-d2 as IS for the analysis of MDA is recommended.

As EBC is composed mainly of water, it can be considered as a rather clean biological matrix. Indeed, we did not observe any matrix effect, as the slope of the calibration curves prepared in EBC and water were statistically identical. This result is consistent with previous data [28,29]. It is thus possible to use calibration curves in water to quantify EBC samples.

To our knowledge, this is the first study that has attempted to characterize the uncertainty associated with the analysis of MDA in EBC. Method validation together with uncertainty measurement provide a way to check whether a method correctly fits for the intended purpose. The measured MDA concentrations in EBC for our healthy, non-smoking workers were ranging between <LOD to 886.2 pg/mL with most of the samples (63%) having concentrations between LOD and LOQ. This stresses the importance of using a sensitive method for MDA analysis in EBC, particularly for healthy volunteers in whom the values can be very small. Our results are comparable but lower than the one reported in the literature (Table 3). Such discrepancy could be due to the absence of blank correction. Whereas the uncertainty at such low concentrations (≈200 pg/mL) might be relatively high, the developed method appears to be sensitive enough to be applicable in epidemiological studies. To our knowledge, only Corradi et al. [47] reported a value attributed to inter-individual variability of MDA in EBC. This value found at 18.5%, was lower than the one we determined in our study (Table 2). For other matrices such as urine, Martinez et al. [48] reported higher variabilities for MDA, with inter-individual variabilities above 300% and intra-individual variabilities of 110%. Additionally, we observed that about 38% of the observed intra-individual could be attributable to the calculated analytical uncertainty. This variability would have a significant impact in studies relying on within-subject differences. This clearly indicates that the quantification of MDA levels in EBC in longitudinal studies is quite challenging, particularly regarding the achievement of a low LOD. Reducing or at least standardizing the blank signal is important to improve the sensitivity of this analytical method.

## 5. Conclusions

In this study, we developed an optimized LC–ESI-MS/MS method for MDA detection and quantification in EBC using DNPH as the derivatizing agent. The developed method presented acceptable performances. The associated expanded uncertainty of this analysis reached a maximum of 23% for MDA concentration close to the LOQ (211 pg/mL). We successfully applied this method to 164 EBC samples from non-smoking workers and demonstrated its accuracy and precision. We propose to use this method in future epidemiological studies focusing on comparing different volunteers or groups of subjects for oxidative stress in the respiratory system.

## Data Availability

The datasets used and/or analyzed besides individual data collected within the ROCoCoP are available from the corresponding author on reasonable request. The latter are not available in accordance with data protection procedures declared to the French ethical commission.

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
