# Peer review of "Method Validation and Characterization of the Associated Uncertainty for Malondialdehyde Quantification in Exhaled Breath Condensate"

_antioxidants, 2021, doi:10.3390/antiox10111661_

Round 1

Reviewer 1 Report

The manuscript presents a thorough work on malondialdehyde quantification including method development and validation with special emphasis on uncertainties characterization. The authors have made valuable corrections to the resubmitted version. I recommend accepting the article after minor corrections.

Abstract: recovery range (92-106.5%) would be more appropriate than the lowest value only

Line 51. “Measurements” (plural) or “has”

Table 1 Shouldn’t the expected performance for the Error for SD 1 = 74 pg. mL-1 be 20% instead of 30%?

Line 337. Probably, it is better to specify these results as outliers for clarity (as it is made in the figure caption).

Lines 382-403. It is still unclear from the text and table, why the method can be considered applicable for use in "epidemiological researches relying on between-subject differences".

Author Response

The manuscript presents a thorough work on malondialdehyde quantification including method development and validation with special emphasis on uncertainties characterization. The authors have made valuable corrections to the resubmitted version. I recommend accepting the article after minor corrections.

1) Abstract: recovery range (92-106.5%) would be more appropriate than the lowest value

Authors’ response: This amendment was performed.

2) only Line 51. “Measurements” (plural) or “has”

Authors’ response: The grammatical error has been corrected.

3) Table 1 Shouldn’t the expected performance for the Error for SD 1 = 74 pg. mL-1 be 20% instead of 30%?

Authors’ response: As the reviewer wisely pointed out, according to the FDA/ICH acceptance criteria, the expected performance at the lowest calibration point is equal to 20%. We made this amendment in the table 1.

4) Line 337. Probably, it is better to specify these results as outliers for clarity (as it is made in the figure caption).

Authors’ response: The manuscript text has been rephrased as suggested by the reviewer. We inserted the term of outliers in the following sentence:  “The acceptance limits of 20 % was exceeded for only two results that can be considered as potential outliers at the concentration of 211.2 pg/mL and the concentration 349.3 pg/mL respectively, confirming the LOQ of 211 pg/mL”. (Lines 343-344)

5) Lines 382-403. It is still unclear from the text and table, why the method can be considered applicable for use in "epidemiological researches relying on between-subject differences".

Authors’ response: We are aware that this part may be difficult to understand considering that the value of inter-individual variability is computed cumulatively with the intra-individual variability. If we assume each point of the intra-variability separately (table 2), the values remains at first sight under the calculated uncertainty of 23%. That is why, to make this assertion less confusing we removed the findings on intra-individual variability and rephrased the abstract as follows:

“Although, the analytical uncertainty was high at low MDA concentrations, it was significantly lower than the observed inter-individual variability”.

We also revised the discussion by deleting the following sentence:

“This analytical uncertainty might have significant effect on the observed intra-individual variability, but might contribute to a smaller extent for inter-individual variability” as this sentence is confusing in the revised version of the document.

Finally, we described more precisely in the results section how the method could be applicable to epidemiological researches relying on between-subject differences. Line 405-412 were rephrased as follows : “

To assess the advantages of our improved analytical method for epidemiological studies, we assumed: (1) a study design where each subject is sampled once; (2) log-normal distributions for MDA concentrations in EBC (as observed in our dataset); (3) a CV of 30% (worst case scenario) corresponding to an intra-individual variance on the log scale of 38 % (intra-subject variability of 50 %), and a total variance of 19 % (inter-subject variability of 50 %). We found the analytical variability to be far lower than the inter-subject variability, and conclude that the proposed analytical method is able to detect relevant inter-subject differences in epidemiological studies.”

Reviewer 2 Report

All comments were addressed by authors. 

Author Response

Thank you very much for your comment.

Reviewer 3 Report

The manuscript “Quantification of malondialdehyde in exhaled breath condensate: method validation and characterization of the associated uncertainty” concerns the optimization and validation of an LC-MS/MS method after derivatization for the determination of an oxidative stress biomarker, malondialdehyde, in exhaled breath condensate. The method was applied to a significant number of samples. The topic of this work relates to human health and safety and it is of high interest, since it concerns a non-invasive method for the determination of a biomarker related to several diseases and can potentially contribute to the analytical tools of such analysis. However, there are some deficiencies of this study that authors are asked to address in order to improve their manuscript.

  1. The concentration of the internal standard (15 ng/mL) is much higher than the concentration range of the analyte (MDA) (74 – 22220 pg/mL). Usually, the concentration of the internal standard is inside the calibration range, in order to have similar sensitivity with the analyte. Authors are asked to comment and justify the selection of the internal standard concentration.
  2. What are the criteria for characterizing the sample as QC? Were they unknown samples with known spiked concentrations?
  3. Figure 2S: the units of x-axis are confusing. If the values concern hours (h) the simple numbers 0, 1, 2 would be preferable instead of 00/00 h, etc…
  4. 4 MDA derivatization with DNPH: why was a temperature of 50°C instead of 37°C based on the data of Figure S2 ? Further elaboration should be provided in the main manuscript on the interpretation of the results. 
  5. Lines 183 – 185: The number of significant figures on the m/z values should be the same for all the m/z. In addition, does the MS resolution of the applied instrument justifies such an accuracy with two decimals in m/z values?
  6. 6 Method validation and estimation of it expanded uncertainty: Please replace “it” with “its”
  7. Table 2: the number of replicates (n=?) based on which the mean values and variability was calculated should be given
  8. Regarding repeatability: It is not clear whether the authors calculated repeatability performing replicates of the same sample or of different samples. In addition to the fact that the Quality Control samples are not reported to be certified samples of known concentration but rather unknown samples with spiked known concentrations, the main text should be clearer on how repeatability was calculated and Table 2 to be more self-explained.
  9. The footnote “a: comprised between LOD and LOQ” in Table 2 is not clear. What does it mean? How were the calculations performed?
  10. Line 199: Please correct “expended”
  11. Lines 278 – 279: Rephrase the sentence. “following” seems not be in the correct place
  12. Table 1 is not comprehensive. It should include the number of replicates for each data set that has been subjected to statistical analysis.
  13. Figure 3: Authors should provide more information, apart from the general relationships reported in Lines 209 – 218, on how the values of these uncertainties have been calculated.

Author Response

The manuscript “Quantification of malondialdehyde in exhaled breath condensate: method validation and characterization of the associated uncertainty” concerns the optimization and validation of an LC-MS/MS method after derivatization for the determination of an oxidative stress biomarker, malondialdehyde, in exhaled breath condensate. The method was applied to a significant number of samples. The topic of this work relates to human health and safety and it is of high interest, since it concerns a non-invasive method for the determination of a biomarker related to several diseases and can potentially contribute to the analytical tools of such analysis. However, there are some deficiencies of this study that authors are asked to address in order to improve their manuscript.

1) The concentration of the internal standard (15 ng/mL) is much higher than the concentration range of the analyte (MDA) (74 – 22220 pg/mL). Usually, the concentration of the internal standard is inside the calibration range, in order to have similar sensitivity with the analyte. Authors are asked to comment and justify the selection of the internal standard concentration.

Authors’ response: This is a very interesting remark. When we carried out the first validation tests, we noticed that the signal of the internal standard MDA-d2 was about 10 times lower than MDA standard with similar concentration. We thus selected a concentration of the internal standard in order that the signal be of comparable intensity than that observed for the MDA calibration standards.  We think that this lowest response for MDA-d2 compared to MDA could be attributable to an incomplete hydrolysis of the 1,1,3,3-tetraethoxypropane-d2 solution. Nevertheless, since the signal difference was stable over time (<0.5% variability), we decided to concentrate the internal standard. In this manner, it would have a similar sensitivity with the analyte and would remain in the same working range. We modified the methods by inserting (lines 127-128) the following sentence “The concentration of the internal standard was selected to be of comparable signal intensity observed for the MDA calibration standards”

2) What are the criteria for characterizing the sample as QC? Were they unknown samples with known spiked concentrations?

Authors’ response: As described in § 2.3., the QC correspond to one matrix (EBC) sample spiked with known MDA concentration. These QC are prepared based on a pooled EBC sample collected from healthy non-smoking volunteers, in order to get the lowest MDA level in this matrix. The basal concentration of this pooled sample is not certified but we determined an average concentration of 98.63 ± 17.11 over twelve independent measurements (table 1). Spiking levels corresponded more or less to calibration levels for standards solution. This approach for QC was selected because certified EBC with certified MDA levels cannot currently be obtained commercially.

More details have been added in the section § 2.3.

3) Figure 2S: the units of x-axis are confusing. If the values concern hours (h) the simple numbers 0, 1, 2 would be preferable instead of 00/00 h, etc…

Authors’ response: The figure S2 has been modified in the supplement information to make it less confusing.

4) MDA derivatization with DNPH: why was a temperature of 50°C instead of 37°C based on the data of Figure S2 ? Further elaboration should be provided in the main manuscript on the interpretation of the results.

Authors’ response: Based on the obtained data (Figure S2), the temperature of 37°C gives the largest signal for the MDA-DNPH but is more variable than at 50°C. Moreover, we observed that derivatization at 37°C was occasionally incomplete. Indeed, MDA-DNPH signal could vary when samples were stored in an auto-sampler at room temperature (23 °C) and reinjected after 24h. This was not observed when samples were derivatized at 50°C. We thus selected an incubation temperature of 50 °C. Details were added in the manuscript lines 271-275.

5) Lines 183 – 185: The number of significant figures on the m/z values should be the same for all the m/z. In addition, does the MS resolution of the applied instrument justifies such an accuracy with two decimals in m/z values?

Authors’ response: Thank you for your relevant comment. The MS resolution of the instrument is indeed limited and does not consider the separation of ions according to their mass-to-charge ratio (m/z) beyond units. The number of significant figures was modified in the text.

6) Method validation and estimation of it expanded uncertainty: Please replace “it” with “its”

Authors’ response: The grammatical error has been corrected.

7) Table 2: the number of replicates (n=?) based on which the mean values and variability was calculated should be given

Authors’ response: As suggested by the reviewer, we added the number of replicates in the table 2.

8) Regarding repeatability: It is not clear whether the authors calculated repeatability performing replicates of the same sample or of different samples. In addition to the fact that the Quality Control samples are not reported to be certified samples of known concentration but rather unknown samples with spiked known concentrations, the main text should be clearer on how repeatability was calculated and Table 2 to be more self-explained.

Authors’ response: As explained in the supplement information, the method repeatability was determined by analyzing in quintuplicate five levels of EBC QC controls (obtained from healthy volunteers and spiked with increasing  concentrations of MDA standards as described in § 2.3 of the manuscript.  Such measurement was repeated in three non-consecutive days over a two-week period and using FDA/ICH guidelines.  On each measurement day, we thawed aliquots of the pooled EBC sample and spiked them with a known MDA concentration. The repeatability for each QC presented in Table 1 corresponds thus to the variability of 3x5=15 measurement. In order to take into account this comment, we added in Table 1 the number of values used to calculate the repeatability and modified the description of the precision calculation in the Supplementary material.

9) The footnote “a: comprised between LOD and LOQ” in Table 2 is not clear. What does it mean? How were the calculations performed?

Authors’ response: These are the concentrations of MDA in volunteers in the range of LOD (70 pg/ml) and LOQ (211 pg/ml) determined during validation. We have changed the footnote to “MDA concentrations were between the LOD (70 pg/ml) and LOQ (211 pg/ml)”

10) Line 199: Please correct “expended”

Authors’ response: The grammatical error has been corrected.

11) Lines 278 – 279: Rephrase the sentence. “following” seems not be in the correct place

Authors’ response: For the clarification of the sentence, the name of the author to whom the manuscript refers has been added in the text.

12) Table 1 is not comprehensive. It should include the number of replicates for each data set that has been subjected to statistical analysis

Authors’ response: As suggested by the reviewer the number of replicates was added for each data set in the table 1.

13) Figure 3: Authors should provide more information, apart from the general relationships reported in Lines 209 – 218, on how the values of these uncertainties have been calculated.

Authors’ response: In order to be as concise as possible in the main manuscript, we added more information in the supplementary information describing how the uncertainty at the LOQ level (as example) was calculated.

Round 2

Reviewer 3 Report

The authors have satisfactorily addressed the reviewer's comments.

This manuscript is a resubmission of an earlier submission. The following is a list of the peer review reports and author responses from that submission.

Round 1

Reviewer 1 Report

MDA is one of the most commonly used biomarkers of oxidative stress (OS). Its concentration can be determined by several methods, including DNPH analysis by LC-MS.

The ms. aimed at characterization and optimization of this approach to quantitate MDA in exhaled breath condensate, where the MDA concentration is low. The authors analyzed 163 EBC samples of different environment and  show that their optimized method is more sensitive, more precise and more reproducible than that  of  other methods  Hence, the authors recommend their optimized  Method for future use in clinical researches 

If I had to determine MDA in exhaled breath,, I would have tried this method

Reviewer 2 Report

This work focuses on development, optimization and validation of the method of malondialdehyde quantification, which is considered to be oxidative stress biomarker, in exhaled breath condensate. This is an interesting topic within the scope of Antioxidants. Being timely and topical issue the article can be interesting to the broad audience of readers. Proper characterization of uncertainties and evaluation of their sources is of key importance for consideration of the applicability of the analytical method for the specific study. The manuscript is well written and presents the thorough comprehensive work done by the authors. I recommend accepting the article after minor revision.

I have some small remarks:

p.1. Line 12 “…  analyzing malondialdehyde (MDA)…” . “Quantification” would probably be more relevant.

p.2 lines 50-54 analysis of the composition of the exhale breath itself, and not only the EBC is worth mentioning here as non-invasive diagnostic approach.

Lines 198-204 and throughout the manuscript: variables given in text looks better being in italics

p.12 Table 3 typo in last column (pg/ml)

p.12 Table 3 (important). The given ref.26 relates to the quantification of MDA in urine and not in EBS using HPLC and not HPLC-MS. So, it is still interesting to compare with (since similar derivatization procedure was applied), but not in the way it was done.

Is the ref.36 correct? I cannot find the values given in the Table in the original article.

Suppl. Page 2. Typo: “The recovery rates was expressed as percent…”

Reviewer 3 Report

The paper is a first in the field attempt at characterizing the uncertainty associated with the analysis of MDA in EBC. To that end, it has merit. The background information could be expanded upon, but it is sufficient in introducing MDA as a by-product of oxidative stress. This said, the authors make it a point to assert its application in epidemiological studies and use it in clinical settings. However, there are several shortcomings from this manuscript, particularly with the assay validation procedures.  The regression model did not use any weighting, which seems very relevant considering the low end of the curve had >30% deviation from expected values, which exceeds all normal guidances (FDA/ICH) that analytical assays follow, and thus makes this assay unacceptable, and the high end of the curve was fine (<1% deviation).  

Abstract:

  • (Line 22): The abstract states “procedural blank subtraction was the most important contributor th analytical uncertainty at LOD levels (reaching 35%).How can the procedural blank even have a calculated uncertainty?

-              “performs sufficiently well relying on within-subject differences.” How is this assertion made if its the first study?

Introduction:

  • (Lines 62-69) discuss prelim analytical studies of MDA using TBA as the coupling acid. Conditions were harsh, and DNPH is more selective for the carbonyl. Plus, MDA-DNPH produces distinct peak.
  • (Lines 81-85) itself brings up the issue of cross reactions with other possible aldehydes
  • (Lines 90-95) are the objectives:
    • Optimize
    • Validate
    • Assess feasibility

Questions

  • (Lines 32-39): discusses MDA as a suitable analyte for pathologies associated with metabolic stress. However, this section does not discuss metabolic diseases due to enzyme deficiency or mutations. Examples include lysosomal storage diseases, G6PD def, etc. Further, the intro discusses oxidized lipids. In which case a discussion of diabetes becomes quite relevant. Lines 45-48 only discuss its application in respiratory pathologies
  • Grammatical error end of (line 51)

methods:

  • Any concerns w/ a d2-MDA w/ crosstalk?
  • What about high QC at 2253 pg/mL, which is > ULOQ @ 2222 pg/mL

Results:

  • (Lines 254-260) discusses residual MDA interfering with signals. However, the paper goes on to explain that DNPH purification did not fix the problem. They had purchased industrial grade DNPH (96% neat) to begin with. I do not follow why the authors thought this would fix the problem.

  • (Lines 289-294) authors state that a linear calibration curve with a regression of >0.995 was determined. However, they also note that at low concentrations the variability is 64.9%. If in fact MDA concentrations is very low in EBC samples, then I am not sure the paper can confidently state their method is robust.
    • (Line 309) Back calculation at low concentrations was >30%, which is unreliable for quantitation, and thus unacceptable as a bioanalytical quantitative assay.
    • (Line 356) , Figure 4, reiterates the point that this method was intended for real world application, but that its uncertainty is above 35% at concentrations of MDA below 100 pg/ml
    • (Line 389) lightly brings up the matter, but does not discuss it any further.

No weighting step was used for linear regression and should be.  The % deviation at low end >23% and at high end was <1%.  Weighting is needed, preferably 1/x2.

This assay should follow FDA/ICH guidelines for bioanalytical assay accuracy and precision of <15% for calibration standards and QCs (and <20% for LLOQ).

It is incorrect to use LOD/2 for samples with a signal below the LOD, i.e. an undetectable peak.  You cannot infer a concentration of LOD/2 when there is no detectable signal for that sample.

Figures 2 and 4 are of poor quality. Since you have R skills, make these figures using ggplot2.

Please present a figure of the EBC study samples, and since the assay was unacceptably imprecise at the low end of the calibrated range, report how many study samples were near the LLOQ.